# Which ethical values underpin England's National Health Service reset of paediatric and maternity services following COVID-19: a rapid review

Anna Chiumento ![ORCID],[1] Paul Baines,[2] Caroline Redhead,[3] Sara Fovargue,[4] Heather Draper,[2] Lucy Frith ![ORCID] [5]

¹Institute of Population Health Sciences, University of Liverpool Faculty of Health and Life Sciences, Liverpool, UK
²Warwick Medical School, University of Warwick, Coventry, UK
³Liverpool Law School, Faculty of Humanities and Social Sciences, University of Liverpool, Liverpool, UK
⁴Law School, Lancaster University, Lancaster, UK
⁵Law and Philosophy, Faculty of Humanities and Social Sciences, University of Liverpool, Liverpool, UK

**Correspondence to**
Dr Anna Chiumento;
Anna.Chiumento@liverpool.ac.uk

## ABSTRACT

**Objective** To identify ethical values guiding decision making in resetting non-COVID-19 paediatric surgery and maternity services in the National Health Service (NHS).

**Design** A rapid review of academic and grey literature sources from 29 April to 31 December 2020, covering non-urgent, non-COVID-19 healthcare. Sources were thematically synthesised against an adapted version of the UK Government's Pandemic Flu Ethical Framework to identify underpinning ethical principles. The strength of normative engagement and the quality of the sources were also assessed.

**Setting** NHS maternity and paediatric surgery services in England.

**Results** Searches conducted 8 September–12 October 2020, and updated in March 2021, identified 48 sources meeting the inclusion criteria. Themes that arose include: staff safety; collaborative working – including mutual dependencies across the healthcare system; reciprocity; and inclusivity in service recovery, for example, by addressing inequalities in service access. Embedded in the theme of staff and patient safety is embracing new ways of working, such as the rapid roll out of telemedicine. On assessment, many sources did not explicitly consider how ethical principles might be applied or balanced against one another. Weaknesses in the policy sources included a lack of public and user involvement and the absence of monitoring and evaluation criteria.

**Conclusions** Our findings suggest that relationality is a prominent ethical principle informing resetting NHS non-COVID-19 paediatric surgery and maternity services. Sources explicitly highlight the ethical importance of seeking to minimise disruption to caring and dependent relationships, while simultaneously attending to public safety. Engagement with ethical principles was *ethics-lite,* with sources mentioning principles in passing rather than explicitly applying them. This leaves decision makers and healthcare professionals without an operationalisable ethical framework to apply to difficult reset decisions and risks inconsistencies in decision making. We recommend further research to confirm or refine the usefulness of the reset phase ethical framework developed through our analysis.

## Strengths and limitations of this study

► The first review to identify the ethical principles guiding decision making in maternity and paediatric services as England's National Health System delivers non-urgent, non-COVID-19 healthcare during the pandemic.

► We conducted a rigorous rapid review of sources from policy, academic and grey literature databases.

► Our approach to qualitative synthesis and appraisal of sources against the Appraisal of Guidelines for Research and Evaluation II (AGREE-II) tool identified areas where ethical guidance and policies lack clarity and fail to implement patient and public involvement.

► Our coding framework is based on the 2017 UK Government Pandemic Flu Ethical Framework, adapted according to two policy sources that met our inclusion criteria, presenting possible methodological tensions.

► An initial Reset Phase Ethical Framework has arisen out of our inductive qualitative synthesis of sources for others to apply and refine.

## INTRODUCTION

The COVID-19 pandemic is causing far-reaching consequences for health systems worldwide. In England, the response to the sudden demand for critical care services was to reorient clinical capacity. Many non-urgent services were suspended, and staff and resources were redeployed to acute care.[1 2] The pandemic's impact on routine healthcare has been severe. For example, in England, a backlog in areas such as cancer diagnosis and elective surgeries accumulated during the first quarter of 2020.[3 4] In April 2020, the UK Government declared that non-COVID-19 clinical services *must* resume alongside the capacity for subsequent waves of COVID-19.[5] This 'reset' of National Health Service (NHS) services encapsulates all the implications

of providing routine care alongside the demands of the COVID-19, including for example, the impacts on caring relationships due to infection prevention and control measures. In this unique 'reset' context, it is unclear which ethical values were underpinning decisions about how to reset health services.[6] Identifying these acknowledge the role of values in policy making,[7] and recognise that decisions that may appear to be based on science, resources or risk are underpinned by value-based judgements.[8–10] To identify which ethical values are underpinning reset decision making in maternity care and paediatric surgery in England, we conducted a rapid review of policy, practice and academic sources.

Our review asked: which ethical values (explicitly or implicitly) guided decision making in non-COVID-19 paediatric surgery (critical/intensive care admissions, surgery, hospital discharge and aftercare) and maternity services (prenatal, intrapartum and postpartum care) during the initial NHS reset in England? We focused on maternity and paediatric services because professional and patient organisations have highlighted adverse impacts on these areas due to measures to respond to COVID-19 infections,[11–14] presenting clear ethical challenges. Maternity services cannot be suspended, and restrictions on accompanying family and carers may have profound effects. We focused on restarting paediatric surgery because of clear ethical conflicts in the suspension of elective paediatric services even though children are, on the whole, relatively unscathed by COVID-19, and because the secondary effects of the pandemic may have a greater impact on children.[15 16]

The pandemic, with emerging evidence and uncertain outcomes, rapid adjustments to healthcare policies and practices—both for the acute and now the reset phase—and uncertainties around personal risk, has created a particularly challenging decision-making context. The ethical values guiding the resumption of non-COVID-19 health services are likely to differ from the everyday ethical frameworks relied on prior to the pandemic. The acute phase of the UK's response to the pandemic has been guided by the Pandemic Flu Ethical framework,[17] which reorients decision making from an individualised to a more public health ethics orientated approach.[18 19] This ethical framing recognises the relational context of decision making,[20] emphasising mutual dependencies. Notably, the pandemic has disproportionately affected certain social groups,[21] including vulnerable older people,[22] those with disabilities[23] and black, Asian and minority ethnic (BAME) communities,[24] thus spotlighting structural inequalities and intersectionalities. It has been proposed that making decisions about healthcare delivery in this context should foreground ethical values such as solidarity,[25 26] reciprocity and fairness. We aimed to identify which ethical values underpinned decisions about how to reset health services in England.[6] This is an important first step in providing an ethical framework for

healthcare professionals and decision makers specific to the reset period[27] and potentially to future pandemics.

## METHODOLOGY

We adopted a rapid review methodology appropriate to addressing urgent demands for synthesised evidence,[28] conducting a qualitative thematic synthesis[29] following the ENTREQ guidelines (30[30] – see completed ENTREQ checklist). The protocol guided a comprehensive yet pragmatic approach to the searches, screening, analysis and appraisal of sources (see online supplemental file 1).

### Inclusion and exclusion criteria

We included sources that: (A) were developed to guide non-COVID-19 paediatric surgery and maternity services, or (B) discussed the application of ethical values to paediatric surgery and maternity services in England during the reset phase. The reset phase commenced on 29 April 2020, the day NHS services were instructed to prepare delivery of non-COVID-19 surgical services,[5] and remains ongoing. Broadly, the reset requires that NHS Trusts:

▶ Resume all non-urgent services incorporating revised COVID-19 infection prevention and control measures.
▶ Prepare for, and manage, second or recurrent waves of COVID-19 infections.
▶ Embrace opportunities to reconfigure health services (eg, accelerating telemedicine).

Accordingly, our inclusion criteria were: sources published after 29 April 2020, relating to non-COVID-19 paediatric and maternity services in the NHS in England, discussing decision making with implicit or explicit reference to ethics and written in English. A cut-off date of 31 December 2020 was introduced when conducting the updated searches in March 2021, as this is when the Health Foundation COVID-19 policy tracker ended. We took an inclusive approach to data sources that met the inclusion criteria if they were national (UK wide and applicable to England), NHS Trust or local policies and directives; guidance or statements from professional bodies; working papers or committee reports; evidence reviews; primary qualitative or quantitative research; peer-reviewed commentaries; or grey literature discussing experiences of paediatric or maternity services in England during the reset phase.

### Electronic search strategy

Searches were conducted between 8 September and 12 October 2020 by AC and PB, and updated between 10 and 21 March 2021 by AC. For academic sources, we searched the bibliographic databases PubMed and PubMed LitCOVID, and clearing houses of COVID-19 related research, including the EPPI Centre Living Map of COVID-19 evidence[31] and Evidence Aid. Recognising the broad scope of our review question, we also searched grey literature sources including websites of UK professional medical bodies (eg, the Academy of Medical Royal Colleges) and clearing houses of COVID-19 sources, such

as the Health Foundation COVID-19 Policy Tracker.[32] Additional grey literature and academic websites identified during the search dates were included in an effort to achieve completeness (eg, ref [33]).

We developed a search strategy (see online supplemental file 1), which was piloted and refined on PubMed (see online supplemental file 2). Where search engines did not facilitate Medical Subject Headings (MeSH) terms, we selected keywords from the list of terms: for example, "paediatric", "maternity", or "COVID-19". For websites where searching was not possible (eg, ref [34]), a manual review of relevant website sections was undertaken. All grey literature search results were documented in Excel spreadsheets or Word documents, and bibliographic database searches in EndNote.

## Publication scheme and freedom of information requests

To complement the electronic searches, we used the Freedom of Information Act 2000 (FOIA[35]) with NHS England Trusts, including those with Clinical Ethics Committees. FOIA imposes two main duties on public authorities: to proactively publish information in a 'publication scheme'[36] and to respond to requests for information. We focused on sources such as policies, decision-making tools, Trust board papers and minutes that detailed approaches to ethical decision-making guiding maternity and paediatric services during the reset period. The publication scheme review addressed two classes of information: '*How we make decisions*' and '*Our policies and procedures*'. Included documents were read in full and coded against the coding framework by CR (see online supplemental file 3). This paper briefly reports a case study example of the publication scheme review.

## Screening

Sources were reviewed and duplicates removed before combining results. All were double screened based on title and abstract, where available. Where unavailable, or when undecided, full-text review was undertaken. AC, PB, LF, CR, CG and SF screened sources, with HD resolving conflicts in double screening decisions. Papers were categorised against a 0–3 scale, where: 0: not included; 1: included – identifies approach to decision making; 2: included – identifies what decision has been made; and 3: included – provides justification for decision(s) taken. Where a source met multiple screening categories, all were identified. This categorisation approach sought to provide an initial sense of the depth of sources to inform full-text analysis. Grey literature screening was conducted in a shared Excel spread sheet and for academic sources using Rayyan software.[37]

## Data analysis

In order to conduct a thematic synthesis of sources, we developed a coding framework for the reset phase. This was based on the Pandemic Flu Ethical Framework[17] adapted according to two interlinked guidance documents: '*Third phase of the NHS response to Covid*', a letter issued by the NHS Chief Executive and Chief Operating Officer to all NHS Trusts,[38] and '*Five Principles for the next phase of the Covid response*', developed by a coalition of UK health and social care charities.[39] The 2017 framework provides a checklist to encourage consideration of the full range of ethical principles in decision-making processes to guide decisions during a pandemic. We adapted the 2017 framework because it was clear that the reset phase may require a different approach to the acute phase. As part of this adaptation, we reduced the Pandemic Flu Ethical Framework (eg, removing the principle of 'flexibility', which was viewed as a subdomain of 'minimising harms and balancing against benefits'), and adjusted subdomains according to how they were operationalised in these two guidance documents (see table 1 for the reset phase coding framework). This adaptation reduced the overlap between principles and subdomains for application as a coding framework. The resulting framework was iteratively refined through data analysis, as described in the results. Inductive coding involved reading each document and coding against the ethical principles and subdomains in the coding framework, alongside a 3–5 line summary of the key points from each document and, where relevant, identifying quotes.

Our approach raises a methodological tension as our coding framework draws on two sources relevant to the review but which were excluded from it. It was, however, justified given the lack of an overarching ethical framework tailored to the reset phase and the need for a coding framework that reflects the ethical specificities of this phase. We will consider this further in the Discussion.

Alongside our thematic synthesis, we assessed the extent to which ethical principles were identified, operationalised and balanced against one another using a 1–3 scale where: (1) ethical principle(s) inferred or mentioned but not clearly applied; (2) ethical principle(s) identified and application described; and (3) ethical principle(s) operationalised, that is, discussed in-depth, including balancing against other principles. This scoring system was an adaptation of our protocol: we had intended to apply the 'review of reasons' approach,[40] but the non-normative nature of the majority of sources rendered this approach unsuitable. Data analysis was led by AC, with PB, CR, SF, LF and CG double coding and scoring 28 sources. Following double coding, the team shared analysis, providing a coding check and discussing emerging findings.

Policy sources (including professional guidance) were appraised for quality using an adapted version of the AGREE-II instrument[41] reduced to seven core questions (see table 2). In selecting the quality appraisal questions, we considered the standards that could be anticipated in guidance for which an evidence base was emerging and where rapid policy and practice decisions were required.[42] Appraisal was conducted independently by AC, PB, SF, CR and CG, drawing on the criteria defined in the AGREE-II Users Manual.[43] This includes scoring of 1–7, where 7:

**Table 1** Reset phase coding framework (adapted from the Ethical Framework in the UK Government's Pandemic Flu Policy[17])

| Ethical principle (from Pandemic Flu Ethical Framework) | Adapted subdomain (based on NHS letter and National Voices Five Principles) |
|---|---|
| Respect | Involvement (ie, right to express views on matters affecting them, engaging those affected by decisions). |
| | Respecting choices about personalised care (best interests of person as a whole). |
| | Collaborative working/engagement (organisational coordination; NHS volunteer scheme, clinical teams, Clinical Commissioning Groups, local authorities; coproduction with voluntary sector, patient orgs, etc). |
| Recognising harms and balancing against benefits (physical, psychological, social and economic) – proportionality | Recover operation of healthcare (including addressing backlog of care needs, resuming home visits for vulnerable/shielding where appropriate). |
| | Safety of NHS staff (physical, psychological, systemic inequalities and flexible working). |
| | Embrace new ways of working (eg, telemedicine, home visits, etc). |
| | Enhance crisis responsiveness (second wave) |
| | Accelerate preventative programmes (obesity reduction, seasonal influenza and outreach to marginalised groups). |
| | Responsiveness (adapt plans to new circumstances/information). |
| Reciprocity | Concept of mutual exchange: take responsibility for own behaviour and reduce others expose others to risks. |
| | Protect those at risk of C19 (physically, socially, BAME, etc). |
| Fairness | Inclusivity in service recovery (eg, barriers or access needs and support those with unequal access to care). |
| | Patient prioritisation (to address backlog that is, clinical urgent /longest waiting, etc). |
| | Reduce health inequalities (social inequalities and social determinants of health). |
| | Everyone matters equally and weighted equally in policies and any disproportionate impact on one particular group is accounted for. |
| Accountability | Transparency (ie, document decisions, clarity of who is responsible for decisions, governance arrangements, assess against milestones and sharing information to help others). |

BAME, black, Asian and minority ethnic; NHS, National Health Service.

strongly agree (the full criteria are met); 2–6: reporting does not meet the full criteria (lacks completeness or quality of reporting); and 1: strongly disagree (no information, poor reporting of the criteria or the authors state that criteria were not met).

### Patient and public involvement
As this was a rapid review, there was no patient or public involvement.

### RESULTS
We present the results of searches, screening, the characteristics of included sources and the data analysis. We also separately present a case study example of the publication scheme review from one NHS Trust. No FOIA responses providing relevant materials were received.

Academic and grey literature searches identified 19 405 sources (10 505 and 8900, respectively). After removing duplicates, 18 766 results were screened, with 18 316 excluded as not relevant. Four hundred and fifty sources were assessed for eligibility by title and abstract or, where necessary, full-text screening. Of these, 360 were excluded as being outside the review scope, and on full-text review a further 39 sources were excluded. Therefore, searches identified 48 sources for analysis (see figure 1).

Table 3 presents key characteristics of the 48 sources, which include professional guidance (n=30) and statements (n=2), government policy statements/letters (n=5), academic papers (n=5), reports of patient engagement (n=2) and of implementing professional guidance (n=1), briefing papers (n=2) and a blog post (n=1). Eighteen sources covered all areas of clinical care, 21 focused on maternity services, 8 on paediatric services and 1 on consent for surgery. The sources covered England or the UK, with some containing Trust-specific case studies. Finally, some sources cross-referenced one another; for example, the Academy of Medical Royal Colleges[44] has accompanying sources focusing on specific areas, such as staff support.[45]

Table 2 summarises the assessment of 42 policy/professional guidance against the AGREE-II tool. Sources scored

**Table 2** AGREE-II assessment of 42 policy guideline sources

| | | AGREE-II questions (domain in brackets) | | | | | | |
|---|---|---|---|---|---|---|---|---|
| Title | Reference | The guideline objective is specifically described (D1) | The guideline development group includes individuals from all relevant professional groups (D2) | The views and preferences of the target population have been sought (D2) | There is an explicit link between the recommendations and the supporting evidence (D3) | Key recommendations are easily identifiable (D4) | The guideline presents monitoring and/or auditing criteria (D5) | Competing interests of the guideline development group members have been recorded and addressed (D6) |
| Principles for reintroducing health services – COVID-19 | 44 | 7 | 5 | 1 | 4 | 7 | 1 | 1 |
| COVID-19. Effects on health from non-COVID-19 conditions and moving forward to deliver healthcare for all | 46 | 6 | 4 | 3 | 3 | 7 | 1 | 1 |
| Preparing for COVID-19 surges and winter | 68 | 7 | 4 | 3 | 3 | 7 | 1 | 1 |
| Reset, restore and recovery: staff support | 45 | 7 | 4 | 1 | 1 | 7 | 1 | 1 |
| Health protection: public and professional responsibilities | 91 | 7 | 4 | 1 | 1 | 7 | 1 | 1 |
| Reset, restore and recovery: medical education and training | 47 | 7 | 4 | 3 | 2 | 7 | 1 | 1 |
| Reset, restore and recovery: equality | 69 | 7 | 4 | 1 | 2 | 7 | 1 | 1 |
| Second phase of NHS response to COVID-19 | 5 | 7 | 5 | 1 | 3 | 7 | 4 | 1 |

**Table 2** Continued

| Title | Reference | AGREE-II questions (domain in brackets) | | | | | | |
|---|---|---|---|---|---|---|---|---|
| | | The guideline objective is specifically described (D1) | The guideline development group includes individuals from all relevant professional groups (D2) | The views and preferences of the target population have been sought (D2) | There is an explicit link between the recommendations and the supporting evidence (D3) | Key recommendations are easily identifiable (D4) | The guideline presents monitoring and/or auditing criteria (D5) | Competing interests of the guideline development group members have been recorded and addressed (D6) |
| Operating framework for urgent and planned services within hospitals: all emergency patients to be tested on admission and elective patients to isolate for 14 days prior to admission | [92] | 2 | 1 | 1 | 1 | 7 | 2 | 1 |
| Second phase of NHS response to COVID-19 for cancer services | [93] | 1 | 3 | 1 | 2 | 5 | 1 | 1 |
| Workforce Race Equality Standard (WRES) briefing for board and COVID-19 emergency preparedness, resilience and response membership in the NHS | [70] | 4 | 1 | 1 | 3 | 5 | 1 | 1 |
| COVID-19: guidance for the remobilisation of services within health and care settings, infection prevention and control recommendations | [94] | 5 | 7 | 1 | 4 | 5 | 1 | 1 |

Continued

**Table 2** Continued

| | Reference | The guideline objective is specifically described (D1) | The guideline development group includes individuals from all relevant professional groups (D2) | The views and preferences of the target population have been sought (D2) | There is an explicit link between the recommendations and the supporting evidence (D3) | Key recommendations are easily identifiable (D4) | The guideline presents monitoring and/or auditing criteria (D5) | Competing interests of the guideline development group members have been recorded and addressed (D6) |
|---|---|---|---|---|---|---|---|---|
| | | AGREE-II questions (domain in brackets) | | | | | | |
| Delivering a paediatric elective surgery service during the COVID-19 pandemic | 95 | 7 | 7 | 7 | 7 | 5 | 3 | 1 |
| COVID-19: guidance for planning paediatric staffing and rotas | 48 | 7 | 1 | 1 | 1 | 6 | 1 | 1 |
| COVID-19 and Us: views from Royal College of Paediatrics and Child Health (RCPCH) and Us | 53 | 7 | 5 | 7 | 7 | 7 | 1 | 1 |
| Ethics framework for use in acute paediatric settings during COVID-19 pandemic | 71 | 7 | 7 | 1 | 5 | 7 | 3 | 1 |
| National guidance for the recovery of elective surgery in children | 58 | 7 | 7 | 5 | 7 | 7 | 4 | 1 |
| Reset, restore, recover – RCPCH principles for recovery | 96 | 7 | 1 | 1 | 3 | 7 | 1 | 1 |

**Table 2** Continued

| Title | Reference | AGREE-II questions (domain in brackets) | | | | | | |
|---|---|---|---|---|---|---|---|---|
| | | The guideline objective is specifically described (D1) | The guideline development group includes individuals from all relevant professional groups (D2) | The views and preferences of the target population have been sought (D2) | There is an explicit link between the recommendations and the supporting evidence (D3) | Key recommendations are easily identifiable (D4) | The guideline presents monitoring and/or auditing criteria (D5) | Competing interests of the guideline development group members have been recorded and addressed (D6) |
| Antenatal care for women without suspected or confirmed COVID-19 and living in a symptom free household | 52 | 5 | 1 | 1 | 5 | 7 | 1 | 1 |
| RCM briefing on reintroduction of visitors to maternity units across the UK during the COVID-19 pandemic | 59 | 4 | 1 | 1 | 3 | 3 | 1 | 1 |
| oyal College of Midwives (RCM) clinical briefing sheet: guidance for midwifery services on 'freebirth' or 'unassisted childbirth' during the COVID-19 pandemic | 72 | 5 | 1 | 1 | 4 | 3 | 1 | 1 |
| Guidance for the provision of midwife-led settings and home birth in the evolving COVID-19 pandemic | 97 | 6 | 6 | 2 | 5 | 4 | 2 | 2 |
| Equality essentials: appropriate risk assessment during the current pandemic | 98 | 5 | 3 | 3 | 3 | 5 | 2 | 2 |

Continued

**Table 2** Continued

| Title | Reference | AGREE-II questions (domain in brackets) | | | | | | |
|---|---|---|---|---|---|---|---|---|
| | | The guideline objective is specifically described (D1) | The guideline development group includes individuals from all relevant professional groups (D2) | The views and preferences of the target population have been sought (D2) | There is an explicit link between the recommendations and the supporting evidence (D3) | Key recommendations are easily identifiable (D4) | The guideline presents monitoring and/or auditing criteria (D5) | Competing interests of the guideline development group members have been recorded and addressed (D6) |
| COVID-19 impact on black, Asian and minority ethnic women | 65 | 6 | 4 | 2 | 5 | 4 | 2 | 1 |
| Principles for the testing and triage of women seeking maternity care in hospital settings during the COVID-19 pandemic: a supplementary framework for maternity healthcare professionals | 99 | 6 | 3 | 2 | 5 | 5 | 3 | 1 |
| Guidance for antenatal and postnatal services in the evolving COVID-19 pandemic | 51 | 6 | 7 | 2 | 5 | 5 | 3 | 1 |
| Antenatal care for women with current suspected or confirmed COVID-19 or with a member of their household with suspected or confirmed COVID-19 | 100 | 6 | 5 | 2 | 6 | 6 | 2 | 1 |

Continued

**Table 2** Continued

| Title | Reference | AGREE-II questions (domain in brackets) | | | | | | | | |
|---|---|---|---|---|---|---|---|---|---|---|
| | | The guideline objective is specifically described (D1) | The guideline development group includes individuals from all relevant professional groups (D2) | The views and preferences of the target population have been sought (D2) | There is an explicit link between the recommendations and the supporting evidence (D3) | Key recommendations are easily identifiable (D4) | The guideline presents monitoring and/or auditing criteria (D5) | Competing interests of the guideline development group members have been recorded and addressed (D6) |
| Domestic abuse: identifying, caring for and supporting women at risk of/victims of domestic abuse during COVID-19 | 66 | 6 | 3 | 3 | 4 | 4 | 2 | 2 |
| Bereavement care in maternity services during COVID-19 pandemic | 74 | 6 | 4 | 6 | 7 | 3 | 1 | 1 |
| Postnatal care for women with suspected or confirmed COVID-19 | 84 | 5 | 7 | 5 | 6 | 4 | 1 | 1 |
| Virtual consultations | 50 | 7 | 5 | 5 | 7 | 6 | 4 | 1 |
| Restarting planned surgery in the context of the COVID-19 pandemic | 56 | 6 | 7 | 1 | 1 | 7 | 1 | 1 |
| Delivering midwifery intrapartum care where local COVID-19 escalation protocols are required to be enacted | 101 | 7 | 5 | 1 | 6 | 5 | 1 | 1 |

Continued

**Table 2** Continued

| Title | Reference | AGREE-II questions (domain in brackets) | | | | | | | |
|---|---|---|---|---|---|---|---|---|---|
| | | The guideline objective is specifically described (D1) | The guideline development group includes individuals from all relevant professional groups (D2) | The views and preferences of the target population have been sought (D2) | There is an explicit link between the recommendations and the supporting evidence (D3) | Key recommendations are easily identifiable (D4) | The guideline presents monitoring and/or auditing criteria (D5) | Competing interests of the guideline development group members have been recorded and addressed (D6) |
| Supporting pregnant women using maternity services during the COVID-19 pandemic: actions for NHS providers | 60 | 2 | 1 | 2 | 2 | 3 | 1 | 1 |
| Important—for action—operational priorities for winter and 2021/2022 | 57 | 5 | 1 | 1 | 1 | 5 | 1 | 1 |
| National Clinical Prioritisation Programme (including evidence based Interventions): frequently asked questions | 67 | 7 | 1 | 1 | 2 | 7 | 1 | 1 |
| Digital by default or digital divide? Virtual healthcare consultations with young people 10–25 years | 55 | 7 | 7 | 7 | 1 | 7 | 1 | 1 |
| Restoring children's health services, COVID-19 and winter planning – position statement | 62 | 1 | 1 | 1 | 3 | 4 | 1 | 1 |

Continued

**Table 2** Continued

| Title | Reference | AGREE-II questions (domain in brackets) | | | | | | | |
| | | The guideline objective is specifically described (D1) | The guideline development group includes individuals from all relevant professional groups (D2) | The views and preferences of the target population have been sought (D2) | There is an explicit link between the recommendations and the supporting evidence (D3) | Key recommendations are easily identifiable (D4) | The guideline presents monitoring and/or auditing criteria (D5) | Competing interests of the guideline development group members have been recorded and addressed (D6) |
|---|---|---|---|---|---|---|---|---|
| Anaesthesia and critical care: guidance for clinical directors on preparation for a possible second surge in COVID-19 | 2 | 1 | 1 | 1 | 1 | 2 | 1 | 1 |
| COVID-19 in pregnancy: information for healthcare professionals | 54 | 6 | 2 | 3 | 4 | 5 | 2 | 2 |
| Joint RCOG and RCM statement: planning for winter 2020/2021 – reducing the impact of COVID-19 on maternity services in the UK | 64 | 7 | 7 | 1 | 7 | 7 | 2 | 1 |
| Midwives call for common sense on maternity visiting guidance | 61 | 7 | 6 | 1 | 7 | 7 | 2 | 1 |

NHS, National Health Service.

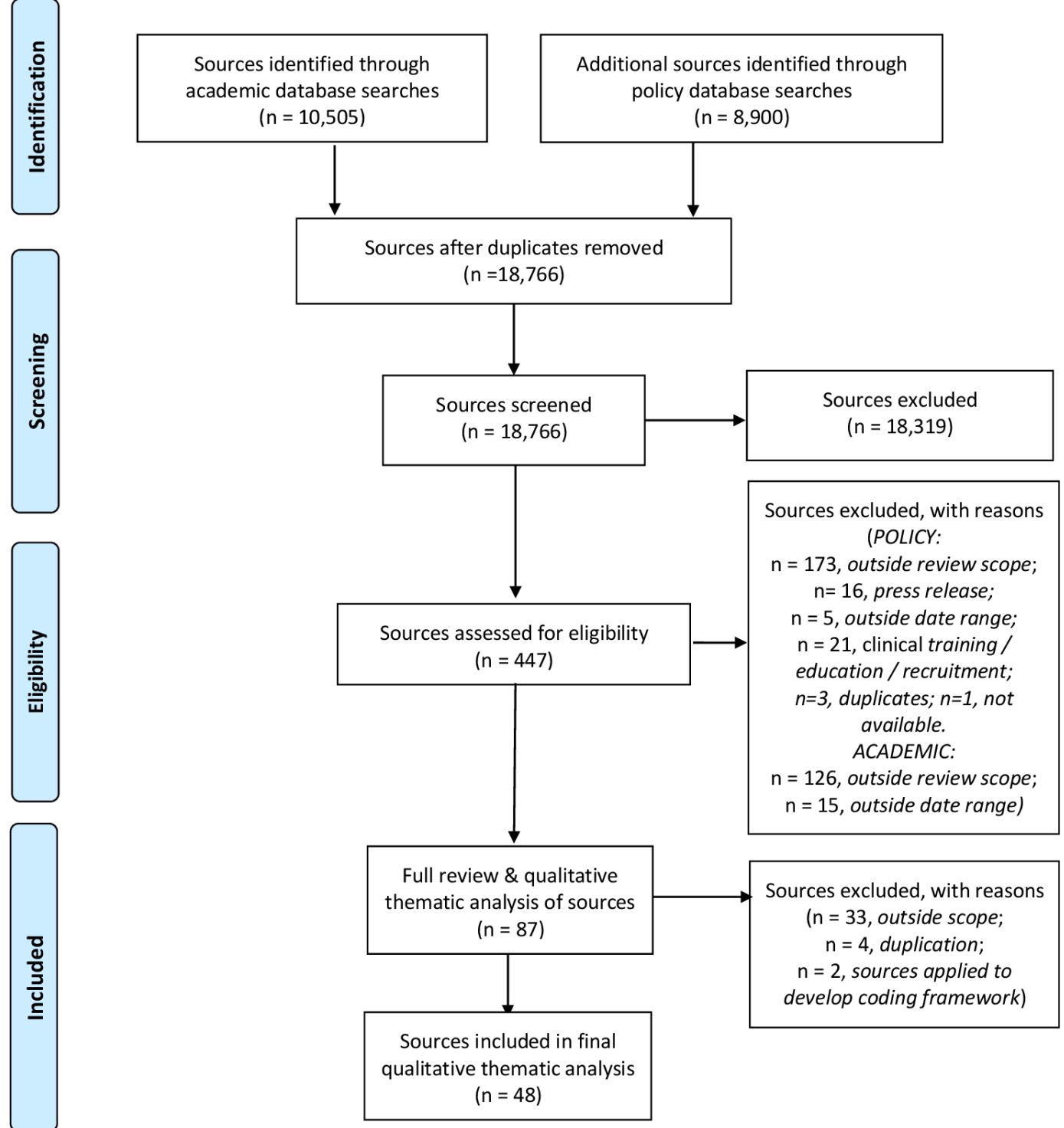

From: Moher D, Liberati A, Tetzlaff J, Altman DG, The PRISMA Group (2009). *Preferred Reporting Items for Systematic Reviews and Meta-Analyses: The PRISMA Statement.* PLoS Med 6(7): e1000097. doi:10.1371/journal.pmed1000097

For more information, visit www.prisma-statement.org.

**Figure 1** PRISMA flow diagram of searches. PRISMA, Preferred Reporting Items for Systematic Reviews and Meta-Analyses.

highest for clarity of the guideline objective (19 scored 7 and 10 scored 6) and easily identifiable key recommendations (19 scored 7). Favourable scores were achieved for the involvement of professional groups (nine scored 7, and 15 between scored 4 and 6). Conversely, low scores were common on seeking views of the target population where 24 sources scored 1, with three scoring 7, and on whether the guideline presented monitoring and/or

**Table 3** Key characteristics of sources

| Title | Reference | Publication type (policy, report, press release, briefing, statement, professional guidance, peer reviewed article, commentary, decision-support tool/ framework, blog) | Date of publication | Population (maternity, paediatrics or all clinical specialities) | Source scope (international, national, regional, trust and hospital) |
|---|---|---|---|---|---|
| Grey literature sources | | | | | |
| Principles for reintroducing health services – COVID-19 | 44 | Professional guidance | May 2020 | All | National |
| Covid-19. Effects on health from non-COVID-19 conditions and moving forward to deliver healthcare for all | 46 | Professional guidance | May 2020 | All | National |
| Preparing for COVID-19 surges and winter | 68 | Professional guidance | Jul 2020 | All | National |
| Reset, restore and recovery: staff support | 45 | Professional guidance | Jun 2020 | All | National |
| Health protection: public and professional responsibilities | 91 | Professional guidance | 11 July 2020 | All | National |
| Reset, restore and recovery: medical education and training | 47 | Professional guidance | June 2020 | All | National |
| Reset, restore and recovery: equality | 69 | Professional guidance | June 2020 | All | National |
| Second phase of NHS response to COVID-19 | 93 | Policy (letter) | 29 April 2020 | All | National |
| Operating framework for urgent and planned services within hospitals: all emergency patients to be tested on admission and elective patients to isolate for 14 days prior to admission | 92 | Policy | 14 May 2020 | All | National |
| Second phase of NHS response to COVID-19 for cancer services | 93 | Policy (letter) | 8 July 2020 | All | National |
| WRES briefing for board and COVID-19 emergency preparedness, resilience and response (EPRR) membership in the NHS | 70 | Briefing | 24 June 2020 | All | National |
| COVID-19: guidance for the remobilisation of services within health and care settings, infection prevention and control recommendations | 94 | Public Health England Guidance | 20 August 2020 | All | National |
| Delivering a paediatric elective surgery service during the COVID-19 pandemic | 95 | Implementation of NICE guidance | 27 July 2020 | All | National |
| COVID-19: guidance for planning paediatric staffing and rotas | 48 | Professional guidance | 10 July 2020 | Paediatrics | National |
| COVID-19 and Us: views from RCPCH and Us | 53 | RCPCH engagement | 4 November 2020 | Paediatrics | National |
| Ethics framework for use in acute paediatric settings during COVID-19 pandemic | 71 | Professional guidance | 1 September 2020 | Paediatrics | National |
| National guidance for the recovery of elective surgery in children | 58 | Professional guidance | 9 November 2020 | Paediatrics | National |

Continued

**Table 3** Continued

| Title | Reference | Publication type (policy, report, press release, briefing, statement, professional guidance, peer reviewed article, commentary, decision-support tool/ framework, blog) | Date of publication | Population (maternity, paediatrics or all clinical specialities) | Source scope (international, national, regional, trust and hospital) |
|---|---|---|---|---|---|
| Reset, restore, recover – RCPCH principles for recovery | 96 | Professional guidance | 19 May 2020 | Paediatrics | National |
| It is right to restart services, but we must do so in a safe way | 49 | Blog | 7 June 2020 | All | National |
| Antenatal care for women without suspected or confirmed COVID-19 and living in a symptom free household | 52 | Professional guidance | 14 August 2020 | Maternity | National |
| RCM briefing on reintroduction of visitors to maternity units across the UK during the COVID-19 pandemic | 59 | Professional guidance | 15 July 2020 | Maternity | National |
| RCM clinical briefing sheet: guidance for midwifery services on 'freebirth' or 'unassisted childbirth' during the COVID-19 pandemic | 72 | Professional guidance | 30 April 2020 | Maternity | National |
| Guidance for the provision of midwife-led settings and home birth in the evolving COVID-19 pandemic | 97 | Professional guidance | 21 October 2020 | Maternity | National |
| Equality essentials: appropriate risk assessment during the current pandemic | 98 | Professional guidance | May 2020 | Maternity | National |
| COVID-19 impact on black, Asian and minority ethnic women | 65 | Professional guidance | 15 July 2020 | Maternity | National |
| Principles for the testing and triage of women seeking maternity care in hospital settings during the COVID-19 pandemic: a supplementary framework for maternity healthcare professionals | 99 | Professional guidance | 10 August 2020 | Maternity | National |
| Guidance for antenatal and postnatal services in the evolving COVID-19 pandemic | 51 | Professional guidance | 19 June 2020 | Maternity | National |
| Antenatal care for women with current suspected or confirmed COVID-19 or with a member of their household with suspected or confirmed COVID-19 | 100 | Professional guidance | 24 July 2020 | Maternity | National |
| Domestic abuse: identifying, caring for and supporting women at risk of/victims of domestic abuse during COVID-19 | 66 | Professional guidance | 13 November 2020 | Maternity | National |
| Bereavement care in maternity services during COVID-19 pandemic | 74 | Professional guidance | 14 July 2020 | Maternity | National |
| Postnatal care for women with suspected or confirmed COVID-19 | 84 | Professional guidance | 14 August 2020 | Maternity | National |

Continued

**Table 3** Continued

| Title | Reference | Publication type (policy, report, press release, briefing, statement, professional guidance, peer reviewed article, commentary, decision-support tool/ framework, blog) | Date of publication | Population (maternity, paediatrics or all clinical specialities) | Source scope (international, national, regional, trust and hospital) |
|---|---|---|---|---|---|
| Virtual consultations | 50 | Professional guidance | 24 July 2020 | Maternity | National |
| Restarting planned surgery in the context of the COVID-19 pandemic | 56 | Professional guidance | 1 May 2020 | All | National |
| Delivering midwifery intrapartum care where local COVID-19 escalation protocols are required to be enacted | 101 | NICE guidance | 20 July 2020 | All | National |
| Supporting pregnant women using maternity services during the COVID-19 pandemic: actions for NHS providers | 60 | Briefing | 14 December 2020 | Maternity | National |
| Important—for action—operational priorities for winter and 2021/2022 | 57 | Policy (letter) | 23 December 2020 | All | National |
| National Clinical Prioritisation Programme (including evidence based interventions): frequently asked questions | 67 | Policy | 23 September 2020 | All | National |
| Digital by default or digital divide? Virtual healthcare consultations with young people 10–25 years | 55 | Report | September 2020 | Paediatrics | National |
| Restoring children's health services, COVID-19 and winter planning – position statement | 62 | Statement | 9 October 2020 | Paediatrics | National |
| Anaesthesia and critical care: guidance for clinical directors on preparation for a possible second surge in COVID-19 | 2 | Professional guidance | 7 October 2020 | All | National |
| COVID-19 in pregnancy: information for healthcare professionals | 54 | Professional guidance | 14 October 2020 | Maternity | National |
| Joint Royal College of Obstetricians and Gynaecologists (RCOG) and Royal College of Midwives (RCM) statement: planning for winter 2020/2021 – reducing the impact of COVID-19 on maternity services in the UK | 64 | Professional guidance | 8 October 2020 | Maternity | National |
| Midwives call for common sense on maternity visiting guidance | 61 | Statement | 15 December 2020 | Maternity | National |
| Academic sources | | | | | |
| Implications for the future of obstetrics and gynaecology following the COVID-19 pandemic: a commentary | 63 | Commentary | | Maternity | National |
| Sustaining quality midwifery care in a pandemic and beyond | 102 | Review article | 25 May 2020 | Maternity | National |

Continued

**Table 3** Continued

| Title | Reference | Date of publication | Publication type (policy, report, press release, briefing, statement, professional guidance, peer reviewed article, commentary, decision-support tool/framework, blog) | Population (maternity, paediatrics or all clinical specialities) | Source scope (international, national, regional, trust and hospital) |
|---|---|---|---|---|---|
| How should surgeons obtain consent during the COVID-19 pandemic? | 103 | 30 June 2020 | BMJ Views and Reviews | All surgery | National |
| Professionally responsible advocacy for women and children first during the COVID-19 pandemic: guidance from World Association of Perinatal Medicine and International Academy of Perinatal Medicine | 76 | 26 November 2020 | Peer-reviewed article | Maternity | International |
| Respectful maternity care in the context of COVID-19: a human rights perspective | 104 | 26 November 2020 | Peer-reviewed article | Maternity | International |

NHS, National Health Service; NICE, National Institute for Health and Care Excellence.

auditing criteria, where 25 sources scored one. When assessing whether there was an explicit link between the recommendations and supporting evidence, 21 scored 1, with only four scoring 7 and one scoring 6 indicating a clear link. Finally, all sources scored one or two for whether the competing interests of members of the guideline development group had been recorded and addressed.

Table 4 summarises the qualitative thematic synthesis of all 48 sources, highlighting the frequency of coding to each subdomain and scores for the operationalisation of ethical principles.

All sources explicitly referenced or applied the principle of recognising harms and balancing these against possible benefits. The subdomain of *safety of NHS staff* was most frequently coded, with *recovering the operation of healthcare* and *embracing new ways of working* explicitly identified slightly less frequently. Staff safety was understood broadly, encompassing Personal Proective Equipment (PPE), testing and isolation protocols, the importance of staff well-being (including leave) and the importance of ongoing staff training.[2 46–49] Concerns about staff training and progression became more prominent as the pandemic continued to cause disruption.[2 45] New ways of working frequently identified telemedicine, an approach that had been effectively applied in remote community maternity care prior to the pandemic.[50] Integrating telemedicine was recommended in the context of trusting relationships built through in-person care,[51] which involved individualised assessments of patients' characteristics and life circumstances,[50] such as the need for interpretation services,[52] and confidentiality concerns.[53] Both maternity and paediatric sources reflected potential risks with virtual care in relation to '*unvoiced concerns*'[54], recommending a low threshold for in-person consultations.[55] In resetting health services, it was anticipated that routine care would resume in a non-linear way[56]; therefore, continuing adaptation to the evolving situation would be required,[2 52] including establishing new '*post-Covid assessment Services*'.[57] To support this, risk management tools and service level models were proposed[2] that accounted for impacts on key areas, such as human resources,[52 58] or sample risk assessments with recommended phases, for example, for reintroducing visitors and sample visiting guidelines.[59 60] Caution against resuming planned healthcare and routine visiting too quickly was advised due to the time and effort required to reorient people and equipment to routine roles and the additional demands of safety and infection control.[49 61] Once re-established, the need to protect routine services from the potential impact of subsequent waves of COVID-19 in the paediatric context was emphasised to avoid further risks to child health as a result of delayed care.[62]

Respect was a frequently explicitly considered principle, encompassing keeping people informed and respecting personal decisions about care, including acknowledging patients' right to express views on matters affecting them both directly and through organisations such as the

| Table 4 | Thematic analysis of sources | |
|---|---|---|
| **Thematic analysis** | | |
| **Principles** | **Subdomains** | **References** |
| Respect | Involvement | 44 46 47 50 51 53–55 58–63 65–68 70–72 74 76 84 92 96 97 99 100 103–105 |
| | Respecting choices about personalised care | 44 50 51 54 55 57 58 60 64 65 71 72 74 76 96 97 100 103 |
| | Collaborative working/engagement | 2 5 44 46 48 51 52 56–59 61–64 66 68 70 72 74 84 92 94–97 101 105 |
| Recognising harms and balancing against benefits (physical, psychological, social and economic) – proportionality | Recover operation of healthcare | 2 5 44–46 48 49 51–54 56 57 59–62 64 66–69 71 74 84 92 95–97 99 100 103–105 |
| | Safety of NHS staff | 2 5 44–51 54 56–66 69 70 76 84 91 92 94–102 105 |
| | Embrace new ways of working | 5 44 50–55 57 58 60 62 63 65 66 68 70 72 74 94–97 99 100 102 104 |
| | Enhance crisis responsiveness | 2 5 44 45 49 58 61 68 71 |
| | Accelerate preventative programmes | 2 5 46 62–66 68 |
| | Responsiveness | 2 48 51–56 58 60–62 66 70–72 74 76 84 96 97 99–101 |
| | *Patient safety* | 2 46 49 51 52 54 55 58 60–62 64–67 71 72 76 91 92 95 97 99 100 102–104 |
| Reciprocity | Mutual exchange | 48 58–61 65 68 91 94 100 |
| | Protect those at risk of COVID-19 | 2 5 44 46 48 51–54 56 58 59 68–70 84 91 92 94 95 97–100 102 |
| Fairness | Inclusivity in service recovery | 2 44 46 50–56 60–66 69 74 84 97 105 |
| | Patient prioritisation | 2 5 44 46 49 56–58 62 65 67 69 71 76 92 |
| | Reduce health inequalities | 50–55 57 60 62–66 69 70 84 96 98 104 |
| | Everyone matters equally | 2 49 54–56 58–63 65 66 70–72 76 99 104 |
| Accountability | Transparency | 5 46 50 53 55 57–62 64 66–72 76 92 94 96 97 |
| | Finance | 5 57 63 |
| | Sustainability | 62 64 |
| **Justification of principles** | | |
| 1 | Principle(s) inferred or mentioned, but not clearly applied | 5 44–46 53 54 57 58 60 61 65 67–69 84 91 92 96 98 100 101 105 |
| 2 | Application of principle(s) described | 47–52 55 59 63 64 66 70 72 74 94 95 99 103 104 |
| 3 | Application of principle(s) discussed in-depth, including balancing against other principle(s) | 2 56 62 71 76 97 102 |

Maternity Voices Partnership.[60] Examples of such involvement included using patients' experiences of lockdown to inform plans for maintaining routine care alongside managing COVID-19.[53] Paediatric sources were notable for high levels of involvement,[53 55] with one including young people's definition of the concept of reset, encompassing '*contact, connections, and interactions with patients*' while accounting for individual needs and circumstances.[62] The use of active public health messaging or outreach to involve patients was also identified[46 58 62 63] and was added to the coding framework as a subdomain of respect.

Collaborative working was explicitly referenced, recognising the codependency of elements of the health service: '*turning on the tap at one end will not necessarily*

*release the flow at the other — there are multiple taps which need to be released in a sequential fashion*'.[46] Embedding collaboration across hospitals and Trusts was called for through local, regional and national coordination, the redeployment of staff across specialities, the accelerated qualification of students and the return of retired staff who had supported human resource capacity during the first wave of COVID-19.[5 46] Over time, the impact of redeployment on the capacity to provide routine services was considered, including the need for some staff to be protected: '*Maternity staff cannot be replaced by other staff groups due to their specialist skill set and protecting this workforce from unnecessary risk is therefore crucial to ensure that maternity care can be sustained*'[64] and protecting routine child health services from adult COVID-19 escalation processes.[62]

Inclusivity in service delivery was emphasised under the principle of fairness. Barriers to maternity care such as English language abilities, immigration status and individualised factors—including risk of domestic abuse or history of human trafficking–were identified.[54 65 66] This subdomain was frequently considered alongside explicit recognition that everyone matters and should be considered equally in policies. For example: '… *it is important to consider the needs of surgical patients on an equal footing with those receiving care for COVID-19 and other medical diseases*'.[56] Sources identified in the updated searches introduced processes for patient prioritisation for elective care[67] and the concept of '*timely and safe discharge*' to maximise the capacity to respond to ongoing waves of COVID-19 infections.[57] Conducting equality impact assessments to ensure rapid adjustments of policies and procedures to address inequalities and meet public duties was also noted.[5 59]

Under the principle of reciprocity, the subdomain of everyone taking actions to protect healthcare workers and patients was explicitly emphasised. Notably, this recognised the increased risks and burdens faced by healthcare staff and those at increased risk of COVID-19 infection and poor outcomes, such as members of BAME communities.[54 68–70] Finally, accountability was implicitly reflected in the subdomain of transparency, with explicit reference to documenting decisions[50 67 71 72] and engaging in monitoring, evaluation[58] and research,[5 68] and calls for continuing data collection and patient involvement to inform policy and decision making.[55] Transparency in governance structures and decision-making processes were also underscored,[3] thereby ensuring adherence to the UK Equalities Act 2010. Sustainability of both NHS resources (such as staffing) and environmental sustainability (notably in relation to disposable PPE) were added to the coding framework as a subdomain emerging from the updated searches.[62 64]

The analysis led to iterative inductive evolution of the coding framework, adding subcategories identified in italics in table 5, which form the ethical framework emerging from this review.

Scoring sources for their practical usefulness to healthcare professionals highlights that nearly half explicitly identified key ethical principles but failed to offer advice about how they might be used in decision making (22 scored 1). Broad statements about core principles were often made, such as respect for patients and minimising harms that were frequently mentioned in relation to infection prevention and control. Nineteen sources scored 2 for clearly identifying ethical principles and suggesting how they might be applied; for example, by identifying decision-making support tools (eg, ref [59]). Seven sources scored three for their focused, practical suggestions regarding the application of the identified ethical principles, often balancing them against one another. For example, the ethical framework for acute paediatric settings[71] balanced treatment prioritisation against resource constraints, identified decision-making tools and engaged with case scenarios to illustrate ethical tensions, such as the disruptions to care pathways for children with complex needs. It is notable that there was no clear correlation between the quality appraisals against the AGREE-II tool and depth of ethical engagement.

## Publication scheme case study

We present initial findings from one NHS Trust publication scheme review (see online supplemental file 4). As with the wider review findings, the Trust board's focus was on patient, staff and visitor safety, including broad concern with the effects of the Trust's decision making on service delivery during the reset period. An example from a maternity service was the creation of a safe space for disclosure of domestic violence by making a small, but important, adjustment to Trust Standard Operating Procedures by adding questions to ask when a pregnant person's partner was not present. This example reflects an awareness of patients' increased exposure to domestic violence as a result of lockdown, demonstrating the benefit of paying attention to ethical considerations including inequality and patient safety in a specific decision-making context.

## DISCUSSION

Our pragmatic rapid review identified the ethical principles referenced in published academic and grey literature and decision-making guidance informing the resetting of NHS paediatric surgery and maternity services. A key review outcome is a reset phase ethical framework inductively developed based on the sources reviewed (table 5). Our results indicate high levels of congruence in the key ethical considerations and areas of ethical tension underpinning the resetting of both maternity and paediatric services. In this discussion, we focus on two areas of ethical distinctiveness in the reset: the ways that relationality was invoked and the emphasis on equity. We also consider the practical usefulness of the included sources for healthcare professionals applying to concrete situations[73] and outline how the reset ethical framework developed through this review might be operationalised.

Relationality was reflected in numerous ways, anchored in the individual and organisational mutual dependencies and responsibilities that have been starkly highlighted by the COVID-19 pandemic. The ethical importance of attending to the adverse impact of the COVID-19 on caring and dependent relationships, seeking to minimise disruption to these as much as possible to meet the needs of patients and family or carers while simultaneously attending to public safety is one example. In our review, the relational context of decision making was prominent, reflecting family and caring relationships inherent to our areas of focus: birthing partners in maternity care, and parents or carers in paediatric services.[60 61 71 74] Explicit steps to minimise harms and maximise staff and patient safety were grounded in risk assessment and infection prevention and control protocols that relied on reciprocal responsibilities. Reciprocity was also explicitly

**Table 5** Reset phase ethical framework inductively developed through the review (adapted from the UK Government's Pandemic Flu Policy Ethical Framework[17])

| Ethical principle (from Pandemic Flu Ethical Framework) | Subdomain |
|---|---|
| Respect | Involvement (ie, right to express views on matters affecting them, engaging those affected by decisions, *active communication/outreach including public health messaging*). |
| | Respecting choices about personalised care (best interests of person as a whole *including decisions in best interests of children and young people*). |
| | Collaborative working/engagement (organisational coordination *including redeployment*; NHS volunteer scheme, clinical teams, CCGs, local authorities, *nightingale and independent hospitals*; coproduction with voluntary sector, patient orgs, *equality, diversity and inclusion of the workforce,* etc). |
| Recognising harms and balancing against benefits (physical, psychological, social and economic) – proportionality | Recover operation of healthcare (including addressing backlog of care needs, resuming home visits for vulnerable /shielding where appropriate; *resources (staffing, spaces and equipment)*. |
| | Safety of NHS staff (physical, psychological, systemic inequalities, flexible working and *meeting staff training needs*). |
| | Embrace new ways of working (eg, telemedicine, home visits, *COVID-19 testing protocols and pathways for low-risk and high-risk care*). |
| | Enhance crisis responsiveness (second wave). |
| | Accelerate preventative programmes (obesity reduction, seasonal influenza, outreach to marginalised groups, *antenatal and postnatal care*). |
| | Responsiveness (adapt plans to new circumstances/information). |
| | *Patient safety (individualised risk protocols and support person/visiting protocols)*. |
| Reciprocity | Concept of mutual exchange: take responsibility for own behaviour and reduce others expose others to risks. |
| | Protect those at risk of COVID-19 (physically, socially, BAME, etc). |
| Fairness | Inclusivity in service recovery (eg, barriers or access needs, support those with unequal access to care). |
| | Patient prioritisation (to address backlog, ie, clinical urgent/longest waiting, *option of continuing to wait and postpone treatment, 'reason to reside' criteria for timely and safe discharge*). |
| | Reduce health inequalities (social inequalities and social determinants of health). |
| | Everyone matters equally and weighted equally in policies and any disproportionate impact on one particular group is accounted for. |
| Accountability | Transparency (ie, document decisions, clarity of who is responsible for decisions, governance arrangements, assess against milestones and sharing information to help others). |
| | *Finance.* |
| | *Sustainability (of NHS services (eg, staffing); environmental sustainability)*. |

BAME, black, Asian and minority ethnic; NHS, National Health Service.

identified in the additional protections for those at risk of adverse outcomes from COVID-19 due to systematic inequalities and intersectionalities.[21 54] The importance of balancing infection prevention and control actions to reduce COVID-19 transmission with other risks to healthcare was explicitly recognised, notably acknowledging the potential emotional impacts for patients attending appointments or giving birth alone. Psychological safety was reflected in explicit calls to attend to the emotional impacts of delivering care during the pandemic and to minimise the risk of staff burnout. Finally, relationality was implicit in interorganisational collaboration locally, regionally and nationally to coordinate continuity of care, emphasising codependencies of different areas of the health service.[75] A distinctive focus on health equity was explicit in sources balancing the needs of those with COVID-19 with those requiring routine healthcare. Health equity was also implicitly reflected in calls for proactive outreach to overcome health inequalities and ensure care was accessed when needed, including public health measures such as immunisation campaigns attending to potential inequalities of access.

Our assessment of the level of engagement with ethical principles found them to be 'ethics-lite'. While key principles were referenced, sometimes only in passing, many sources failed to operationalise them. We define operationalisation as applying ethical principles to specific situations, considering how predictable ethical dilemmas might be managed or offering suggestions as to how, in practice, ethical principles might be balanced against one another. This is especially important when the ethical approach moves between individual-focussed clinical care and wider public health measures, which is recognised to produce a '*jarring and unwelcome*' (p. 871) shift in ethical framing that clinicians must negotiate.[76] In recognising this, we do not call for prescriptive guidance for every circumstance, rather that guidance should inform and constrain the judgements of those applying them.[73] To achieve this, *how* they ought to be operationalised must be clear. Guidance lacking this dimension leave healthcare professionals without a coherent ethical framework to support decision making,[27] which can result in moral distress.[77] Moreover, '*Research in psychology has demonstrated that when people are working in stressful situations under pressure of time, with access to extensive yet conflicting information from multiple sources, and when outcomes are uncertain, they tend to make more decisions based on intuition, gut feelings, or heuristics (rules of thumb) rather than on rational thinking (Kahneman, 2011)*' (p. 2).[63] This exactly describes the COVID-19 context, with emerging evidence and uncertain outcomes, rapid adjustments to healthcare policies and practices—both for the acute and the reset phase—and uncertainties around personal risk. In such situations, consistently interpreting and applying broad-brush ethical guidance to practice becomes impossible. A clear ethical framework to underpin decision making is therefore required.[73 78]

Our reset ethical framework, inductively developed through this review, offers a useful starting point. Additional research is required to confirm or further refine its congruence with the decision-making processes of individual Trusts and healthcare providers, embedded within their regional and systemic relationships and to areas of healthcare beyond paediatric surgery and maternity services. This forms part of our ongoing research activities. Recognising the importance of our review finding that ethical frameworks should be operationalisable, we briefly explain how our reset ethical framework could be applied in practice. The Pandemic Flu Ethical Framework emphasises *equal concern and respect* as the underpinning principle,[79] which is echoed in our review where *fairness*, chiefly that *everyone matters equally and is weighted equally*, has emerged as an underpinning principle. However, our review demonstrates that the NHS operational context in the reset is ethically distinct. The underpinning principle of fairness must be balanced across considerations such as the impact of delayed care; constraints of infection prevention and control measures; broad mutual interdependencies between healthcare providers, patients and the public; and uncertain COVID-19 risks—exacerbated by inequalities and intersectionalities—for healthcare providers and patients. These considerations foreground complex, layered configurations of interdependencies and relationships embedded within healthcare provision in the reset. Ethical frameworks may assist decision makers to navigate this challenging decision-making context. Consequently, in contrast to the UK Chief Medical Officers advice not to produce updated ethical guidance for the COVID-19 pandemic,[80] our review indicates that the ethically distinctive COVID-19 healthcare operational context *urgently* requires a tailored approach.[81] We agree with the Scottish Government[82] that such a framework should be operationalised to support organisational and individual-level decision making at national, regional and local levels; for example, through Trust specification (see, eg, ref[83]) and with the pragmatic advice and consultation of Clinical Ethics Committees and, where relevant, patient involvement groups.

Appraising sources against the AGREE-II tool identified a lack of monitoring and auditing systems for rapidly adjusted policies and practice guidance, which is concerning given the reported impacts on some areas of patient care. It also showed a lack of public involvement beyond, at best, patient representatives,[84] and a lack of transparency around potential competing interests in guidance development. The government's phase 2 letter provided Trusts the short timeline of 21 weeks to design their service reset.[5] Engagement processes, already time consuming, had to be adapted to online formats. It is, therefore, not surprising that public involvement was lacking. However, in March 2020, NHS England restated the statutory, and ethical, duty to maintain public involvement in decisions about service provision,[85] suggesting that this should have taken place. Public involvement is fundamental to public trust in the collective actions of the NHS and the standards of professional ethical practice of individual healthcare providers.[86–88] This is essential to meet the NHS Constitution's guiding principle, that '*the NHS is accountable to the public, communities and patients that it serves*'.[89] As such, public and patient involvement provides an important moral foundation for difficult ethical decisions in the reset phase and beyond.[90]

Our review maintained methodological rigour by including a systematic search strategy where possible and double screening and double coding 25% of sources. Team discussions to develop the coding framework and reflect on emerging findings were also ongoing throughout. We adopted an inclusive approach to grey literature and academic sources, ensuring the relevance of our review to healthcare policy and practice. This was complemented by the publication scheme review, which indicated the application of guidelines to situated Trust-level decision making. However, methodological limitations remain, chiefly that the rapidity of the review rapidity necessarily limited its scope and depth[42] and may not have identified all relevant sources. Time constraints prevented a multiple appraisal of policy sources as recommended by the AGREE-II tool.[43] Where double coding

arose as a result of a source being revised and included in updated searches, some discrepancies arose in AGREE-II appraisals, which were managed by awarding the highest scores. Time constraints also meant that only CR analysed the publication scheme data. A key methodological challenge in this review was the tension in developing the coding framework from two sources that met the review inclusion criteria. We believe this was acceptable given the inductive and iterative thematic synthesis approach, which led to the inductive development of a revised framework that reflects the distinctive considerations facing decision makers and clinicians during the reset phase. Finally, the breadth of our review question made the adoption of approaches designed for normative reviews challenging and resulted in the use of a scoring system that accommodated our review scope.

This review has sought to render explicit the ethical values underpinning decision making specific to the reset phase, yielding important learning for healthcare policy makers and Trust decision makers. Our findings suggest that some key ethical and legal duties—such as involvement–have been immediate casualties of the time-pressured decision-making context. We accept that there may be significant logistical barriers to achieving meaningful engagement and that compromises during a crisis may be required.[17] However, we recommend that guidance is transparent about any lack of involvement and the reasons for this, while seeking to re-establish meaningful engagement as quickly as possible. We are encouraged that updated searches identified increased involvement of patients, notably informing the resumption of paediatric services[62] and promoting the role of patient representative organisations such as the Maternity Voices Partnership.[60] We also recommend that those developing policy and practice guidance pay attention to their practical application. This will ensure that any normative decision making is operationalisable in the context in which healthcare professionals are working.

## CONCLUSION

This review adds to the rapidly evolving evidence on England's health systems' response to the COVID-19 pandemic, focusing on the normative foundations underpinning the resetting of NHS health services in maternity and paediatric surgery services, alongside a continuing response to the demands of COVID-19. It is important that the government and professional bodies continue to engage with the difficult ethical decisions this requires, and we recommend increased public involvement in this process to build solidarity in supporting the required responses. Our review has found that to date, guidance developed for this period are ethics lite and fail to provide an operationalisable ethical framework for decision makers and healthcare professionals. Addressing this is an important priority as the NHS in England moves further into the reset period, where difficult ethical decisions about *how* health services resets will continue to be

necessary. We are supporting this process by publishing our proposed reset ethics framework here. This has been inductively developed based on the sources included in this review. We continue to refine this framework through our ongoing empirical and conceptual research.

**Acknowledgements** The authors would like to thank Dr Rui Hill, University of Liverpool for his helpful guidance on early stages of developing the review protocol, suggestions for review software, and support with writing up the review search strategy. We would also like to thank Dr Diego Silva, Melbourne University, for his feedback on an earlier draft of this paper as discussant at an Australasian Association of Bioethics in Health and Law workshop in December 2020; and the helpful reflections of other members of the workshop session. Finally, our thanks to CG, University of Liverpool, who contributed to the data analysis, quality assessment and data interpretation of the sources included in the updated March 2021 searches.

**Contributors** LF, HD, AC, SF and PB designed the rapid review concept, question and protocol. AC, LF, HD, SF, PB and CR were involved in various stages of conducting the review, as specified in the paper. All authors were involved in regular team meetings to discuss and reflect upon review conduct and emerging findings. AC led the writing of the paper, with all authors providing review and feedback, and approving the final version for publication.

**Funding** This rapid review is the first phase of the Everyday and Pandemic Ethics project (https://www.liverpool.ac.uk/population-health-sciences/departments/health-services-research/key-projects/resetethics/) funded by the UKRI AHRC Covid-19 rapid response call (AH/V00820X/1).

**Competing interests** None declared.

**Patient consent for publication** Not required.

**Provenance and peer review** Not commissioned; externally peer reviewed.

**Data availability statement** All data relevant to the study are included in the article or uploaded as supplementary information. Additional data available on reasonable request to the corresponding author.

**ORCID iDs**
Anna Chiumento http://orcid.org/0000-0002-0526-0173
Lucy Frith http://orcid.org/0000-0002-8506-0699

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
