## [Reviewer comments · BMJ Open]

ARTICLE DETAILS

TITLE (PROVISIONAL)	Which ethical values underpin England's National Health Service reset of paediatric and maternity services following COVID-19: a rapid review.
AUTHORS	Chiumento, Anna; Baines, Paul; Redhead, Caroline; Fovargue, Sara; Draper, Heather; Frith, Lucy

VERSION 1 – REVIEW

REVIEWER	Taylor, Julie University of Birmingham, School of Nursing, College of Medical and Dental Sciences,
REVIEW RETURNED	22-Feb-2021

GENERAL COMMENTS	Thank you for your rigorous rapid review on the ethical values underpinning England's National Health Service reset of paediatric and maternity services following Covid-19. I really liked the paper and have only a few small suggestions for enhancement. For those for whom ethical principles are unfamiliar or hazy in memory, it might be helpful to add a couple more sentences to explain the importance of This created a unique 'reset' context in which it is critical to consider which ethical values should underpin decisions about how to reset health services. In what way is this critical? Why is there an assumption (that I agree with BTW!) that ethical decision should underpin these decisions, rather than (say) resources. You provide more detail of this in the methods section, but I wonder if might work better as background. In your 1-3 scale of the ethical principles, why was there not a zero score for those sources where there was no apparent consideration at all? Or was this not a feature in any? Use of the AGREE II was also useful, but I would have liked to have been clearer (without constant cross-checking) as to which sources scored high on AGREE but not on ethics, or vice versa (and so forth). Is there a way for this to be presented (in a table for eg?), and perhaps some further discussion. Whilst you acknowledge the methodological tension in adapting two sources into one framework, you have actually justified this well and it is very helpful. Overall the emergent framework is extremely helpful and I hope will enhance decision making in these difficult times. Thank you.
---

REVIEWER	Mullins, E
-----------------	------------

	Imperial College London, metabolism, digestion and reproduction
REVIEW RETURNED	02-Mar-2021

GENERAL COMMENTS	Thank you for this article. The concept is excellent and considers how the NHS should ethically reset services. I think the adapted framework could be useful and should be put out for rapid, wider stakeholder comment with a view to use in service resets. I have two major areas of concern which I would be grateful for the authors comment on. 1. Subject matter and clinical involvement in the study. It would be interesting to know why maternity and paediatric surgery were considered, of themselves and together. I note there are no obstetricians or paediatric surgeons on the author list who might have questioned this combination. Maternity services have continued through the pandemic and adapted, rather than paused, to the circumstances. As such, whilst service delivery models have changed, there will be no reset as such. Issues around the risks and benefits of having partners present for different encounters with maternity care, coming into hospital and using telemedicine have arisen and need to be addressed. Paediatric surgery is considered as a whole, although emergency or time-critical surgery (which represents a significant proportion of the workload e.g. cardiac surgery in babies with congenital heart disease), has continued through the pandemic and elective services have been paused. Results for the different services are presented together. Were there any notable differences between them? There is a Maternity Voices Partnership in most UK maternity trusts for patient engagement which can be activated relatively rapidly and I wonder if this would impact aspects of the Respect domain in comparison with paediatric surgery. 2. Scope. The planned inclusion of national guidance and policy documents and documents produced by NHS trusts is a strength of the study. However, practically I am unclear how the authors have approached all e.g. 190+ UK maternity trusts and a similar number which provide paediatric surgery. It is surprising that no FOI requests for this information from trusts were forthcoming, but unsurprising that this would be low yield in the midst of the pandemic. In general the conclusions are well articulated but could be presented in
---

VERSION 1 – AUTHOR RESPONSE

REVIEWER 1: Prof. Julie Taylor, University of Birmingham, Birmingham Women’s and Children’s Hospital NHS Foundation Trust.

Thank you for your rigorous rapid review on the ethical values underpinning England’s National Health Service reset of paediatric and maternity services following Covid-19. I really liked the paper and have only a few small suggestions for enhancement.

Response: Many thanks for these positive comments about the paper.

For those for whom ethical principles are unfamiliar or hazy in memory, it might be helpful to add a couple more sentences to explain the importance of This created a unique 'reset' context in which it is critical to consider which ethical values should underpin decisions about how to reset health services. In what way is this critical? Why is there an assumption (that I agree with BTW!) that ethical decision should underpin these decisions, rather than (say) resources. You provide more detail of this in the methods section, but I wonder if might work better as background.

Response: We thank the reviewer for her helpful comments and observations around strengthening our focus on ethical principles as crucial for underpinning reset decisions. In light of this feedback, and similar observations from reviewer 2, we have added an additional line and accompanying references to the introductory paragraph to foreground how decision-making is underpinned by ethical or value-based judgements: "In this unique 'reset' context it is unclear which ethical values were underpinning decisions about how to reset health services. Identifying these acknowledge the role of values in policy-making, and recognises that decisions that may appear to be based upon science, resources, or risk are underpinned by value-based judgements" (p.2, lines 27-31). We hope that this addresses the reviewers concern to foreground why we are moving beyond issues of resource allocation, to explore the broader values underpinning a range of decisions.

In your 1-3 scale of the ethical principles, why was there not a zero score for those sources where there was no apparent consideration at all? Or was this not a feature in any?

Response: Thanks to the reviewer for this observation. We have reviewed the use of this scale and would like to draw attention to the 0 having been used at the screening stage to indicate that a source was to be excluded for no engagement with ethical principles (see p.4, line 28-30). As such, for the analysis stage we knew that all included sources had some level of ethical engagement, and therefore applied the 1-3 scale to assess the depth of ethical engagement.

Use of the AGREE II was also useful, but I would have liked to have been clearer (without constant cross-checking) as to which sources scored high on AGREE but not on ethics, or vice versa (and so forth). Is there a way for this to be presented (in a table for eg?), and perhaps some further discussion.

Response: Thank you to the reviewer for this observation about the potential correlation between ethical engagement and quality appraisal of the sources. As a study team we discussed this suggestion, and felt that whilst the AGREE-II assessment was intended to provide a quality check on the sources, we would not necessarily anticipate a correlation between scores on AGREE-II and depth of ethical engagement. However, we recognise that a similar observation may be made by other readers, and have addressed this with a sentence in the results section: "It is notable that there is no clear correlation between the quality appraisals against the AGREE-II tool, and depth of ethical engagement." (p.22, line 12-13).

Whilst you acknowledge the methodological tension in adapting two sources into one framework, you have actually justified this well and it is very helpful. Overall the emergent framework is extremely helpful and I hope will enhance decision making in these difficult times. Thank you.

Response: We thank the reviewer for her encouraging comments, and hope that this framework provides the foundations for wider engagement and refinement to aid clinical decision-makers in the particularly challenging situations they are currently facing.

REVIEWER 2: Dr. E Mullins, Imperial College London

Comments to the Author:

Thank you for this article. The concept is excellent and considers how the NHS should ethically reset services. I think the adapted framework could be useful and should be put out for rapid, wider stakeholder comment with a view to use in service resets.

Response: Many thanks to the reviewer for his encouraging overall assessment of the paper and its relevance to the NHS reset. We agree that wider stakeholder engagement and comment would be an important next step, with a view to piloting use of the framework to consider its role in informing ongoing reset decision-making.

I have two major areas of concern which I would be grateful for the authors comment on.

1. Subject matter and clinical involvement in the study. It would be interesting to know why maternity and paediatric surgery were considered, of themselves and together. I note there are no obstetricians or paediatric surgeons on the author list who might have questioned this combination. Maternity services have continued through the pandemic and adapted, rather than paused, to the circumstances. As such, whilst service delivery models have changed, there will be no reset as such. Issues around the risks and benefits of having partners present for different encounters with maternity care, coming into hospital and using telemedicine have arisen and need to be addressed. Paediatric surgery is considered as a whole, although emergency or time-critical surgery (which represents a significant proportion of the workload e.g. cardiac surgery in babies with congenital heart disease), has continued through the pandemic and elective services have been paused.

Results for the different services are presented together. Were there any notable differences between them? There is a Maternity Voices Partnership in most UK maternity trusts for patient engagement which can be activated relatively rapidly and I wonder if this would impact aspects of the Respect domain in comparison with paediatric surgery.

Response: We thank the reviewer for these reflections on the scope of the review and its presentation. We address each in turn:

- We acknowledge the reviewer highlighting the operationalisation of “reset”. We have added a line stating: “This ‘reset’ of NHS services encapsulates all the implications of providing routine care alongside the demands of the coronavirus, including for example the impacts upon caring relationships due to infection prevention and control measures.” (p.2, line 24-27). We also note that as the reviewer indicates, some aspects of Maternity services were fully suspended – notably visiting (p. 2<="" span="" style="font-family: Calibri;">40-41) – but also water births and in some cases home births. For the paediatric context, we state that our interest is in the ethical conflicts of suspending elective paediatric surgery services in light of the secondary impacts this may lead to for children (p. 2, line 41-44), and therefore the ethical tensions this presents. In our description of the inclusion/exclusion criteria we further differentiate the reset from restarting, emphasising balancing between Covid and non-covid-19 services. We hope this captures our operationalisation of “reset” and thank the reviewer for encouraging additional clarity here.
- Relatedly, our interest in paediatric surgery and maternity care recognises that it would be unfeasible to cover all areas of clinical care. As such we selected these two areas of care as case study examples that are united in their commonalities in caring and dependent relationships. This is highlighted in the introduction for maternity care: “...restrictions on accompanying family and carers may have profound effects” (p.2, line 40-41); and picked-up in the discussion: “In our review, the relational context of decision-making was prominent, reflecting family and caring relationships inherent to our areas of focus: birthing partners in maternity care, and parents or carers in paediatric services” (p.22, lines 40-42). We also bring to the reviewers attention that PB was a paediatric consultant, and that LF and HD have longstanding research interests in ethical issues arising in maternity care. As such our selection of these areas of care draws upon the knowledge and experience team members have of these two areas.
- We also thank the reviewer for encouraging greater engagement with the focus on these two areas of care. To address this, we have sought to more clearly draw out the commonalities between the findings for maternity and paediatric surgery in the results

section. For example, we note that in relation to telemedicine: “Both maternity and paediatric sources reflected potential risks with virtual care in relation to “unvoiced concerns”, recommending a low threshold for in-person consultations.” (p.19, line 12-14). Similarly we highlight calls from both Maternity and Paediatric services to protect staff and resources from Covid-19 escalation processes to ensure routine services continue: “Over time, the impact of redeployment on the capacity to provide routine services was considered, including the need for some staff to be protected: "Maternity staff cannot be replaced by other staff groups due to their specialist skill set and protecting this workforce from unnecessary risk is therefore crucial to ensure that maternity care can be sustained", and protecting routine child health services from adult COVID-19 escalation processes” (p.19, line 44 – p.20, line 2). In terms of involvement we have been struck by the high levels of involvement in paediatric services: “Paediatric sources were notable for high levels of involvement, with one including young people’s definition of the concept of reset, encompassing “contact, connections, and interactions with patients” whilst accounting for individual needs and circumstances” (p.19, line 32-35). Finally, we have added a line to the discussion to emphasise the high levels of commonalities in key ethical principles and concerns across these 2 areas of care: “Our results indicate high levels of congruence in the key ethical considerations and areas of ethical tension underpinning the resetting of both maternity and paediatric services.” (p.22, line 28-30). We hope these additions addresses the reviewers observations.

2. Scope. The planned inclusion of national guidance and policy documents and documents produced by NHS trusts is a strength of the study. However, practically I am unclear how the authors have approached all e.g. 190+ UK maternity trusts and a similar number which provide paediatric surgery. It is surprising that no FOI requests for this information from trusts were forthcoming, but unsurprising that this would be low yield in the midst of the pandemic.

Response: We thank the author for recognising the strengths of our broad inclusion criteria for this review. To clarify the methodological approach: we did not approach individually all maternity trusts and paediatric surgery units in England. Instead, we used broad policy databases - such as the Academy of Medical Royal Colleges, NICE website, or the Covid-hub - which act as repositories of resources and include trust-specific documents as examples. This approach is described under “electronic search strategy” (p.3-4). We also note in our paper the opportunities presented by conducting a publication scheme review and FOI requests to complement the higher-level documentation we have been able to access (p.4, lines 14-23), and would recommend integrating this approach into future reviews.

VERSION 2 – REVIEW

REVIEWER	Taylor, Julie University of Birmingham, School of Nursing, College of Medical and Dental Sciences,
REVIEW RETURNED	04-May-2021

GENERAL COMMENTS	Super paper, will be an important contribution.
---

REVIEWER	Mullins, E Imperial College London, metabolism, digestion and reproduction
REVIEW RETURNED	16-May-2021

GENERAL COMMENTS	Thank you for your responses and edits.
---